# Surgical Site Infiltration with Comfort-in Device and Traditional Syringe in Dogs Undergoing Regional Mastectomy: Evaluation of Intra- and Postoperative Pain and Oxidative Stress

**DOI:** 10.3390/ani14131902

**Published:** 2024-06-27

**Authors:** Giovanna Lucrezia Costa, Fabio Bruno, Fabio Leonardi, Patrizia Licata, Francesco Macrì, Rocío Fernández Parra, Giuseppe Bruschetta, Vincenzo Nava, Michela Pugliese, Filippo Spadola

**Affiliations:** 1Department of Veterinary Sciences, University of Messina, Via Palatucci Annunziata, 98168 Messina, Italy; fabio.bruno@unime.it (F.B.); plicata@unime.it (P.L.); fmacri@unime.it (F.M.); giuseppe.bruschetta@unime.it (G.B.); vnava@unime.it (V.N.); michela.pugliese@unime.it (M.P.); filippo.spadola@unime.it (F.S.); 2Department of Veterinary Science, University of Parma, Via del Taglio 10, 43126 Parma, Italy; 3Departamento de Cirugía y Medicina Animal, Universidad Catòlica de Valencia San Vicente Màrtir, 46018 Valencia, Spain; rocio.f.parra@ucv.es

**Keywords:** surgical site infiltration, Comfort-in device, oxidative status, pain, mastectomy

## Abstract

**Simple Summary:**

The surgical site infiltration of a local anesthetic, defined as the direct injection of a drug into the surgical field, has proven to be effective in the management of intra- and postoperative pain as it allows one to reduce the use of additional analgesics, such as opioids and non-steroidal anti-inflammatory drugs. Tissue infiltration with local anesthetics is usually performed using a traditional syringe attached to a needle. Innovative devices such as Comfort-in^®^ have been developed, and they do not require a needle. This technology is an injection system that works by infusion via a spring system. It releases the drug at high speed, creating a “fluid needle” that penetrates tissue in less than 1/3 of a second at an average pressure of 3900 pounds per square inch (psi). The depth of penetration depends on the volume of the drug, and the dispersion of the drug is uniform through nebulization. In human medicine, local anesthesia with a needle-free injection system has been used in infiltrations, incisions, dental extractions, urology, diabetology, virology, dermatology, and pediatrics. The needle-free injection system has recently been proposed as a valid alternative method also in veterinary medicine for vaccination and drug administration in swine.

**Abstract:**

The surgical site infiltration of a local anesthetic is defined as the direct injection of a drug. This study aimed to compare the effects of surgical site infiltration with 4 mg kg^−1^ lidocaine using a Comfort-in device and traditional syringe on oxidative status and intra- and postoperative pain in dogs undergoing regional mastectomy. Sixty adult female dogs divided into C (Comfort-in device), S (traditional syringe), and CTR (control) groups received 2 µg kg^−1^ dexmedetomidine and 4 mg kg^−1^ tramadol IM, 5 mg kg^−1^ tiletamine/zolazepam IV, and isoflurane. The physiological and anesthesiological parameters were measured. The assessment of intra- and postoperative responses to the surgical stimulus was performed using a cumulative pain scale (CPS score of 0–4) and the Colorado Pain Scale (CSU-CAPS score of 0–4). The hematological and biochemical parameters and inflammatory oxidative status were measured. The CPS scores showed no significant differences between the C and S groups (*p* = 0.236), while the comparison between the CTR, C, and S groups, respectively, showed a significant difference (*p* < 0.001). The postoperative analgesia scores were significantly lower in the C group compared to those of the S and CTR groups (*p* < 0.001). In the C group, no subject received rescue analgesia during the intra- and postoperative periods. The level of oxidative inflammatory stress was lower in group C than those in S and CTR groups, and no side effects were observed in all the groups.

## 1. Introduction

Malignant mammary tumors are the most common form of cancer in female dogs, with an incidence rate of approximately 90%. Most breast tumors are hormone-dependent; therefore, they can be prevented with ovariectomy, which can be performed in the same operating session as mastectomy, before the tumor is removed to avoid the dissemination of neoplastic cells in the abdominal cavity. Mastectomy can be partial nodulectomy when the neoplastic mass is less than 5 mm, circumscribed by a capsule, and superficial to the mammary gland. Simple mastectomy is the excision of the entire malignant mammary gland. Regional mastectomy consists of the excision of the affected breast and of the adjacent ones also affected by neoplastic masses. Regional mastectomy can be bilateral; the latter is usually performed in two surgical sessions. General anesthesia associated with local anesthesia is the most suitable anesthetic for mastectomy for the management of postoperative pain [1].

The incisional infiltration of a local anesthetic is defined as the direct injection of a drug into the surgical field, and it is a technique used to support multimodal analgesia. It is used due to its relative simplicity, safety, and low cost [2,3,4]. The administration of a local anesthetic in combination with a general anesthetic is commonly performed in many protocols for balanced anesthesia, allowing them to work on patients undergoing major surgery with the use of fewer analgesic drugs (such as opioids) and general anesthetics [5,6]. Local anesthetics, such as lidocaine, bupivacaine, levobupivacaine, and ropivacaine, perform their primary analgesic action by preventing the generation and transmission of nerve impulses by blocking the influx of sodium across the cell membrane of nerve axons and inhibiting the conduction of action potentials [7].

The technique of incisional infiltration with local anesthetics has proven to be effective in managing intra- and postoperative pain, reducing the use of additional analgesics [2,8,9,10,11]. Surgical site infiltration is usually performed using a traditional syringe attached to a needle; nevertheless, needle-free innovative devices have been developed, such as the Comfort-in^®^. This technology is a needle-free injection system that works as a spring-loaded infusion system. It releases the drug at high speed, creating a “fluidic needle” that penetrates the skin in less than 1/3 of a second at an average pressure of 3900 pound-force per square inch (psi) [12]. The depth of penetration depends on the volume of the infused drug, and the range of penetration varies from 2.1 to 6.1 mm depending on the kind of employed device (soft, medium, or normal). The spray dispersion of the drug occurs uniformly and subcutaneously with the “sputtering” method. In human medicine, the local infiltration of anesthesia with a needle-free injection system has been employed in dental extractions, urology, diabetology, virology, dermatology, and pediatrics [12,13,14,15,16]. The needle-free injection system has recently been proposed as a valid alternative method for vaccination and drug administration in swine [17]. Human studies have demonstrated that postoperative pain can become chronic. Indeed, the sensitization of the central nervous system (CNS), even in the absence of excessive tissue damage during mastectomy, can lead to the onset of chronic non-specific pain [18,19]. The CNS sensitization caused by a continuous nociceptive stimulus may lead to the amplification of pain perception, resulting in hypersensitivity to the painful stimulus [20]. The International Association for the Study of Pain (IASP) has recently defined this alteration of nociception as “nociplastic pain” [21]. Surgery gives rise to acute nociception, increasing the number of reactive oxygen species (ROS) [22]. The formation of ROS could be the main cause of chronic pain [23]. The surgical technique and the anesthetic protocol used can influence inflammatory oxidative stress due to surgery. Therefore, the evaluation of innovative anesthetic and surgical techniques should be performed other than with cumulative scales, such as the CPS and subjective score scales [24,25,26], also with objective methods, such as the determination of lipid peroxidation (MDA), catalase (CAT), myeloperoxidase (MPO), superoxide dismutase (SOD), and butyrylcholinesterase (BuChe), that increase during lipid peroxidation and inflammation [27,28,29].

This study aimed to compare the influences of surgical incisional line infiltration with traditional syringes and Comfort-in device on analgesia and oxidative status in dogs undergoing partial mastectomy.

## 2. Materials and Methods

This study was approved by the University of Messina Ethics Committee (No. 021/2018). This clinical study was performed in accordance with Italian law (D.M. 116192) and European law (O.J. of E.C. L 358/1 12/18/1986). Software G*Power 3.1 was used to adequately determine the sample size. An “a priori” ANOVA test (fixed-effects, omnibus, one-way) was conducted, with an effect size (f) of 0.45, significance level (α) of 0.05, power (1 − β) of 0.85, and three groups. The animals enrolled in this study were 60 adult female mixed-breed dogs, weighing 10 ± 0.2 kg and aged between 8 and 10 years, with a healthy status as verified by clinical history, physical examination, and blood analysis. Informed consent was obtained from each dog’s owner. The inclusion criterion was the need to perform partial caudal mastectomy due to the presence of a circumscribed neoplastic mass of less than 5 mm. The exclusion criterion was the dogs had not previously been subjected to ovariohysterectomy. The subjects were divided into the following three groups: group CTR (control, 20 subjects), group S (syringe, 20 subjects), and group C (Comfort-in™, 20 subjects). Division was random and established through a draw.

### 2.1. Treatment Administration and Anesthesia

The female dogs received the last dose of food and water eight hours before the start of surgery. The subjects were premedicated with the intramuscular (IM) injection of 2 µg kg^−^^1^ dexmedetomidine (Dexdomitor 500 µg mL^−^^1^, Orion, Milan, Italy) and 4 mg kg^−^^1^ tramadol (Altadol 50 mg mL^−^^1^, Formenti, Milan, Italy) mixed in the same syringe. When a good level of sedation was achieved, catheterization of the cephalic vein was performed for the administration of drugs and fluids (Ringer’s lactate solution, Novaselect, Potenza, Italy). All the dogs received 5 mg kg^−^^1^ tiletamine/zolazepam (Virbac, Carros, Cedex, France) intravenously (IV) and were intubated using a cuffed endotracheal tube which was connected to a circle non-rebreathing system (Mapleson B in parallel of LacK). The patients received isoflurane (IsoFlo, ESTEVE, Milan, Italy) and 100% oxygen during spontaneous breathing for the maintenance of anesthesia. Ten minutes before surgery, the incision line was infiltrated with 4 mg kg^−^^1^ lidocaine; (Lidocaine 2%, Zoetis, Barcellone, Spain). Infiltration was performed in the C group using the Comfort-in™ technology (nozzle capacity 0.5 mL—GamasTECH, Catania, Italy) and in the S group using an insulin syringe (capacity 1 mL − 27 G × 1/2″ − 0.4 × 13 mm, Benefis s.r.l., Genova, Italy). The CTR group did not receive lidocaine in the incision line.

### 2.2. Physiological and Anesthesiological Parameters Measured

On surgery day, the female dogs were weighed using a veterinary scale (ARW EOS, Milano, Italy), and they were acclimatized for 20 min in the pre-surgical room. The baseline parameters were recorded as follows, heart rate (HR) using a stethoscope, respiratory rate (RR) counting thoracic excursion for 1 min, non-invasive arterial blood pressure (NIBSP) (systolic (SAP), diastolic (DAP), and mean (MAP)), and body rectal temperature (T, °C), using a multiparametric monitor (AMI Italia s.r.l., Leonardo model, Napoli, Italy). After premedication at the induction of anesthesia, mastectomy, and skin suturing, the HR, RR, and non-invasive arterial blood pressure were recorded. End-tidal carbon dioxide tension (EtCO_2_, mmHg), the inspired contraction of sevoflurane (ETiso, mmHg), and capillary oxygen hemoglobin saturation (SpO_2_, %) were recorded after induction under general anesthesia during mastectomy and skin suturing. Body rectal temperature (T, °C) was measured also post-surgery. The aforementioned parameters, after premedication, were recorded using a multiparametric monitor (AMI Italia S.R.L., Leonardo model, Napoli, Italy). At the end of skin suturing, isoflurane and oxygen were no longer administered.

### 2.3. Assessment of Intra- and Postoperative Responses to Surgical Stimulus

The evaluation of responses to the surgical stimulus was performed by assigning scores from 0 to 4 to the percent changes in HR, RR, and SAP using the values recorded after general anesthesia, at the beginning of mastectomy, and when suturing the skin. To assign the score, the following scheme was used: 0 ≤ 0%; 1 ≥ 0%, but ≤ 10%; 2 ≥ 10%, but ≤ 20%; 3 = > 20%, but ≤ 30%; and 4 = ≥ 30% or more. The sum of the three scores was the total score. The cut-off point for intraoperative analgesia was scored ≤ 10, which corresponds to an increase of 20–30% in the three aforementioned parameters. If the above score exceeded this, a rescue bolus of 2 µg kg^−1^ of fentanyl was given to the dog (Fentadon 50 µg mL^−1^ Dechra, Torino, Italy) [24,25,26]. A postoperative pain score from 0 to 4 was assigned, from awakening, three hours after administration of lidocaine, and every 6 h until 24 h after surgery using the Colorado State University Veterinary Medical Center (Canine Acute Pain Scale). Point 2, corresponding to moderate or mild pain, was the cut-off point for the administration of postoperative rescue analgesia consisting of 0.2 mg kg^−^^1^ methadone IM (Semfortan 10 mg mL^−^^1^, Eurovet Animal Health B.V. Handelsweg, Bladel, The Netherlands). The scores were assigned by three independent observers unaware of the drug treatment received by the patients.

### 2.4. Partial Mastectomy

The surgical area was trichotomized and made aseptic. The patients were placed in dorsal recumbency. Ten minutes before surgery, the incision line was infiltrated with 2 mL of lidocaine, distributed in 4 applications of 0.5 mL, at a distance of one centimeter from the neoplastic mass. To perform mastectomy, an elliptical incision was made on the last three caudal mammary glands one centimeter from the neoplastic mass and the surrounding macroscopically healthy breast tissue. The subcutaneous tissue was dissected from the abdominal fascia using blunt-tipped scissors, and the superficial epigastric vessels located near the inguinal ring were cauterized and simultaneously cut using a bipolar electrosurgical unit (Valleylab^®^, Covidien FT10, Laguna Niguel, USA). The edges of the skin were brought to the center of the wound, with stitches placed under the skin and sutured. The subcutis and skin were sutured with detached knot sutures using absorbable synthetic materials made of Vicryl 3–0 and 2–0 polyglycolic acid (Ethicon^®^, Livorno, Italy). The surgeries was performed by the same surgeon [1].

### 2.5. Hematological and Biochemical Parameters and Inflammatory Oxidative Stress

The cephalic vein was used to collect 6 mL of blood to carry out the measurements developed by the same operator. Blood samples were divided into two portions, one of which was transferred to a Serum Clot Activator Z Vacuette^®^ (Greiner Bio-One, Kremsmünster, Austria). This Vacuette^®^ was used to measure various biochemical parameters, including aspartate transaminase (AST) and alanine aminotransferase (ALT) activities, glycemia, albumin, BUN, total protein, and oxidative status by measuring lipid peroxidation (MDA), catalase (CAT), myeloperoxidase (MPO), superoxide dismutase (SOD), and butyrylcholinesterase (BuChe). The other aliquot of blood sample was placed in a K3-EDTA Vacuum tube, Vacuette^®^ (Greiner Bio-One, Kremsmünster, Austria), and IDEXX Italy (Milano) assessed the blood count (only for the baseline). The tubes were promptly stored at 4 °C, and the samples were processed (centrifuged at 1500× *g*, 15 min) to achieve a serum aliquot. Blood samples were taken at the baseline, 12, and 24 h after the completion of partial mastectomy. Glycemia was measured using a glucose oxidase/peroxidase assay, total protein using the biuret procedure, and albumin using the bromocresol green technique. The kinetic methodology was used to measure AST and ALT levels at 37 °C, and all the parameters were studied using a UV-Vis A560 spectrophotometer (Fulltech, Rome, Italy) [30,31]. Lipid peroxidation (MDA), catalase (CAT), superoxide dismutase (SOD), myeloperoxidase (MPO), and butyrylcholinesterase (BuChe) were measured to determine the oxidative stress levels.

### 2.6. Determination of Malondialdehyde

Phosphoric acid (85%, 15 mol/L), sodium hydroxide, SDS (8.1%), and sodium chloride were bought from Merck (Darmstadt, Germany). Thiobarbituric acid (TBA) was purchased from Fluka (Buchs, Switzerland). All of the reagents were of analytical grade or the highest grade available. Malondialdehyde (MDA) standard was prepared through the hydrolysis of TMP; TBA reagent (0.11 mol/L: 800 mg TBA dissolved in 50 mL, 0.1 mol/lNaOH) was prepared for the assay. For the quantitative determination of TBARS, 200 μL of an MDA standard solution was used instead of plasma. MDA stock solutions were prepared through the hydrolysis of 50 μL of TMP (10 mmol/L) in 10 mL 0.01 M hydrochloric acid for 10 min at room temperature. MDA stock solution was diluted with ultrapure water to different concentrations of MDA standards. Calibration of the plasma was achieved by adding phosphoric acid containing different amounts of MDA to the pooled samples of plasma. After adding the samples (200 μL) to the reaction mixture, the tubes were incubated at 90 °C in a water bath. Lipid peroxidation of the blood serum samples was evaluated by measuring the production of MDA after reacting with glacial acetic acid thiobarbituric acid for 1 h. The tubes were then placed into ice to arrest the reaction. After cooling to room temperature, 100 μL of standards and samples was placed into a flat-bottom 96-well multitier plate. Absorption was read at 535 nm and 572 nm to correct for baseline absorption with the plate reader. MDA equivalents (TBARS) were calculated using the difference in absorption at the two wavelengths, and quantification was determined with the aid of calibration curves [32].

### 2.7. Determination of Oxidative Parameters

The activity of CAT was established through an enzyme reaction in H_2_O_2_. All the reagents were left at 20 °C. The final volume for each well was 240 μL, and each sample was assayed in duplicate with the use of catalase buffer (11×) and diluted assay buffer. Absorbance measurements were obtained at 540 nm using a BIORAD 680 microplate reader (BIORAD Laboratories, Italy) [33]. Superoxide dismutases (SODs) are catalytic proteins useful for the decomposition of the O_2_^−^ into O_2_ and H_2_O_2_. For each test performed in duplicate, a total volume of 230 μL was determined, with all the reagents used at 25 °C, except xanthine oxidase, and absorbance was monitored between 440 and 460 nm using the BIORAD 680 plate reader (BIORAD Laboratories, Italy) [34]. MPO activity was measured using the dianisidine-H70 method with a 96-well plate, and the samples were transformed in triplicate into a mixture consisting of 0.53 mM o-dianisidine HCl, 0.15 mM H_2_O_2_, and 50 mM K_3_PO_4_ buffer (pH 6.0). After incubation for 5 min at 25 °C, the reaction was stopped by the addition of 30% NaN_3_, and absorbance was measured at 460 nm using a BIORAD 680 plate reader (BIORAD Laboratories, Italy) [35]. The BuChE assay is based on the formation of thiocholine by the hydrolysis of a substrate and a subsequent reaction with 2-nitrobenzoic acid dissolved in 625 µL BuChE Assay Buffer. Absorbance was immediately measured at 412 nm at 25 °C using a BIORAD 680 plate reader (BIORAD Laboratories, Italy) [36].

### 2.8. Statistical Analysis

Statistical analysis was performed using SPSS 27.0 (IBM Company, Italy). A Shapiro–Wilk normality test was conducted. The data are expressed with mean ± SD, median, and range, as appropriate. Two-way ANOVA for repeated measures, followed by Bonferroni test and Friedman test, as appropriate, were used to evaluate the changes along the timeline and differences among groups. Inter-observer agreement was measured using Kendall’s concordance coefficient (W). SPSS automatically corrects nonparametric data, except scores, by transforming them into their base 10 logarithms. *p* < 0.05 was considered significant.

## 3. Results

All 60 female dogs completed this study. The actual power was 0.85. The inter-observer agreement was high (W = 1). The data are not normally distributed. The heart rate showed no significant differences within the groups and among the groups (*p* > 0.05). The respiratory rate showed no significant differences along the timelines in the C and S groups, respectively. The respiratory rate in the CTR group showed a significant decrease compared to the baseline at induction and maintenance under general anesthesia (*p* < 0.001). There are no significant differences among the groups regarding the respiratory rate (*p* > 0.819). The systolic, diastolic, and mean pressures did not show significant differences along the timeline and among the groups (*p* > 0.05). End-tidal carbon dioxide tension (EtCO_2_) did not show significant differences along the timeline and among the groups (*p* > 0.05). The vaporizer was set to 3% of isoflurane in all the groups. The ETiso was 2.3/1.5 mmHg in the C and S groups (*p* = 0.8). In the CTR group, the ETiso was higher compared to those of the C and S groups 3/2.5 mmHg (*p* > 0.05) (details not shown in Table 1). Capillary oxygen hemoglobin saturation (SpO_2_) showed values of 98/100% among the groups The body rectal temperature decreased along the timeline in the three groups (*p* < 0.001), and there were no significant differences in body temperature among the groups (Table 1).

The Cumulative Pain Score (CPS) was below the cut-off point in the C group, and not one of the subjects in this group received rescue analgesia during surgery (Table 2). In the S group, only one subject received rescue analgesia during suturing, while in the CTR group, only one subject received rescue analgesia during mastectomy, and four subjects received rescue analgesia during suturing.

In the C and CTR groups, the CPS scores showed no differences along the timeline (*p* = 0.052), while in the S group, the CPS scores showed differences along the timeline (*p* = 0.000). The comparison of the CPS scores between the C and S groups showed no significant differences (*p* = 0.236), while the comparison between the CTR group and the C and S groups showed a significant difference (*p* = 0.000). The postoperative analgesia scores according to the Colorado Pain Score (CSU-CAPS) showed a significant difference along the timeline in the three groups (*p* = 0.000). In the C group, no subject received rescue analgesia during the postoperative period, while in the S and CTR groups starting from 12 h after surgery, two and four subjects, respectively, required the administration of rescue analgesic. There was no significant difference in the number of dogs requiring intra- and postoperative rescue analgesia among the treatments. The postoperative analgesia scores were statistically lower in the C group than those in the S and CTR groups (*p* = 0.000), while no significant differences in the postoperative analgesia scores (*p* = 0.802) were recorded between the CTR and S groups (Table 3).

Biochemical parameters: aspartate transaminase (AST), alanine aminotransferase (ALT), albumin, BUN, and total protein showed no significant differences either along the timeline or among the groups. A significant reduction in albumin was observed in all three groups at 12 h after surgery. However, at the baseline, the CTR group showed a significantly higher level of albumin than those in the C and S groups (Table 4). In addition, a significant increase in glycemia was observed at 12 h after surgery from the baseline in all the groups. At the baseline, the comparison of glycemia among the groups showed significant differences between C and CTR (*p* = 0.007) (Table 4).

The analysis of malondialdehyde (MDA) levels in the CTR and S groups revealed significant differences from the baseline (*p* < 0.0001). The MDA concentrations in group C did not show significant differences from the baseline (*p* > 0.9999). No significant statistical differences were recorded between the CTR and S groups at 24 and 48 h (*p* > 0.9999). The comparison between the CTR and the C groups showed significant differences over time (*p* = 0.0015, *p* = 0.0007). The comparison between the S and the C groups showed significant differences over time (*p* = 0.0008, *p* = 0.007). (Table 5). In the CTR group, the catalase levels (CAT) showed a significant difference with the baseline across the timeline (*p* = 0.033). The comparison of the CAT levels at 24 and 48 h showed a statistically significant difference (*p* = 0.0228). In the S and C groups, the CAT levels did not show statistically significant differences along the timeline (*p* > 0.05), In group C, the CAT levels at 24 and 48 h showed a statistically significant difference (*p* = 0.0412). There were no significant statistical differences in the CAT levels at the baseline, 24, and 48 h between the groups (*p* > 0.9999) (Table 5). No statistically significant differences were found in the level of oxidative parameters, SOD, MPO, and BuChE, and even along the timeline at 24 and 48 h between the groups (*p* > 0.9999) (Table 5).

## 4. Discussion

In the C group, the Comfort-in technology was used, and the intra- and postoperative administration of rescue analgesia was not required. While in the S and CTR groups, some subjects received rescue analgesic both intra- and postoperatively, and this may have influenced the pain scores measured at the next time point; in fact, the CPS scores between groups C and S did not show significant differences. The isoflurane requirements were similar in groups C and S and greater in the control group than those in the groups that received lidocaine at the incision site. There was probably no difference in the duration of the block between groups C and S, only the better diffusion of lidocaine and the absence of trauma due to the needle with Comfort-in technology. Similar results were reported from a clinical study conducted on humans, in which odontostomatological interventions were performed with the administration of a local anesthetic using a syringe without a needle and with a traditional syringe; no obvious differences emerged in the quality of analgesia obtained. However, the patients treated with the needle-free syringe showed a greater level of comfort when administering the local anesthetic [37].

The determination of oxidative state may represent an objective evaluation method for comparing surgical techniques, anesthetic protocols, and anesthetic drug administration techniques as in this study [38]. The concentration of MDA indicates the degree of oxidative stress [29,39]. A redox imbalance can also modify the plasma concentrations of enzymes, such as catalase (CAT), superoxide dismutase (SOD), myeloperoxidase (MPO), and butyrylcholinesterase (BuChe).

The MDA levels increased at 24 and 48 h in the three groups. However, in the C group, the increase in MDA levels did not show significant differences between the baseline and 24 and 48 h, respectively, and also between 24 and 48 h. This may be due to the better distribution of lidocaine with the Comfort-in method, which produced better durability, and the absence of needle trauma.

No statistically significant differences were found in the levels of the oxidative parameters, SOD, MPO, and BuChE, even along the timeline at the baseline, at 24 and 48 h, and among the groups. The increase in capillary permeability due to post-surgical inflammation can lead to a reduction in the level of albumin in the blood [40]. Hyperglycemia may favor the development of post-surgical pathologies, and therefore the determination of perioperative glycemia is recommended to evaluate the risk of postoperative complications, especially in elderly patients and those with diabetes. However, glycemia is not a specific parameter indicative of surgical stress [40,41,42].

The sensitization of the central nervous system, even in the absence of excessive tissue damage, by a nociceptive stimulus can lead to the amplification of pain perception with hypersensitivity to the painful stimulus [43]. This type of altered nociception was recently defined by the International Association for the Study of Pain (IASP) as nociplastic pain [44]. Every surgical intervention induces inflammatory and oxidative stress [28]. Inflammation induced by surgery can slow down the healing process of damaged tissues [28]. Surgery gives rise to acute nociception, resulting in an increased number of reactive oxygen species (ROS) [22]. ROS formation could be the main cause of chronic pain [23]. Research is increasingly aimed at identifying surgical and anesthetic techniques able to reduce the inflammatory stress phenomena linked to surgical interventions [28].

The surgical removal of tumors can increase the risk of recurrence. Surgical stress, resulting in inflammation and pain, has been shown to increase the risk of metastasis. The anesthesia chosen to perform surgery on the cancer patient could influence the evolution of neoplastic disease. Some studies have shown that regional anesthesia could reduce the risk of metastasis formation. The peritumoral infiltration of lidocaine before surgery at the incision site prevents the formation of metastases, since local anesthetics block the sodium channels [44,45].

The determination of plasma concentrations of MDA and oxidative parameters such as glycemia and albuminemia can be valid biomarkers for the objective evaluation of a surgical or anesthetic technique as in this study [7,46].

Furthermore, Comfort-in may favor the use of a lower dose of local anesthetic for tissue infiltration [12]. In awake animals, the Comfort-in technology can find a valid clinical application because it reduces trauma, and the needle-free injection technique has already been used for the vaccination of pigs [17].

One of the limitations of the present study was that the methods employed for the attribution of pain scores were not sensitive enough for this type of study. To counterbalance this limitation, we supported the results with the determination of oxidative inflammatory stress. Another limitation was not being able to determine the duration of the sensory block with objective methods. A further limitation of this study was that the patients were monitored for analgesia for only 24 h, which corresponded to the duration of hospitalization after surgery.

## 5. Conclusions

In the patients on which we used the Comfort-in technology, the intra- and postoperative use of rescue analgesia was not required. The oxidative parameters did not show significant changes compared to the baseline. Furthermore, no side effects were detected.

## Figures and Tables

**Table 1 animals-14-01902-t001:** Influence of 2 µg kg^−^^1^ dexmedetomidine, 4 mg kg^−^^1^ tramadol, 5 mg kg^−^^1^ tiletamine/zolazepam, and 4 mg kg^−^^1^ isoflurane and lidocaine in surgical incision line infiltration with a traditional syringe (S) and Comfort-in device (C) on physiological parameters compared to control (CTR).

Groups	Baseline	Premedication	Induction	Mastectomy	Suturing Skin
**HR (beats min) ^p^**					
**CTR**	140 ± 13.0	109.5 ± 11.6	107.5 ± 5.7	121 ± 6.2	111.5 ± 4.8
**S**	140 ± 13.2	108 ± 11.8 *	108 ± 5.0 *	104.5 ± 3.9 *	110.5 ± 5.6
**C**	132 ± 6.5	92 ± 16.6	100 ± 14.3	89 ± 15.0	91.5 ± 4.9
**RR (breaths min) ^p^**					
**CTR**	30 ± 1.5	13.5 ± 2.6 *	12 ± 1.5 *	28.5 ± 4.4 *	30 ± 1.4
**S**	28 ± 1.5	13.5 ± 2.5 *	12 ± 1.4 *	12.5 ± 1.2 *	33.5 ± 2.7
**C**	26 ± 3.4	20 ± 2.6	16.5 ± 2.2	16.5 ± 2.5	24 ± 3.1
**SAP (mmHg)**					
**CTR**	165 ± 16.1	142 ± 24.9	143.3 ± 22.9	159.8 ± 25.7	177.8 ± 19.3
**S**	164.9 ± 17.7	141.7 ± 24.7	143.5 ± 24.3	144.6 ± 26.89	157.4 ± 15.4
**C**	159.7 ± 24.7	150.3 ± 28.4	158.6 ± 26.5	154.6 ± 33.0	146 ± 17.5
**MAP (mmHg)**					
**CTR**	122.6 ± 22.9	116.1 ± 27.7	106.2 ± 13.9	123.6 ± 19.8	130.5 ± 16.7
**S**	122.3 ± 22.9	115.2 ± 27.1	105.8 ± 13.7	105.9 ± 15.4	126.1 ± 16.5
**C**	116.1 ± 22.9	110.2 ± 29.1	119.2 ± 16.0	115 ± 22.9	116.8 ± 15.7
**DAP (mmHg)**					
**CTR**	94.4 ± 26.9	83.9 ± 22.3	83.4 ± 22.4	101.1 ± 20.6	105.3 ± 21.8
**S**	94.4 ± 26.9	83.9 ± 22.3	83.4 ± 22.4	88.1 ± 15.1	103.6 ± 11.4
**C**	92.2 ± 27.6	90.1 ± 22.4	100.3 ± 15.9	93.9 ± 23.9	92.5 ± 13.7
**Etco_2_ (mmHg)**					
**CTR**			39 ± 3.6	35.67 ± 1.2	33 ± 3.6
**S**			35.7 ± 1.2	35.3 ± 1.5	36.33 ± 1.2
**C**			35.7 ± 1.2	34.7 ± 2.5	33 ± 3.6
**T (°C)**					
**CTR**	38.51 ± 0.2				36.9 ± 0.4 *
**S**	38.32 ± 0.2				36.9 ± 0.3 *
**C**	38.58 ± 0.5				36.9 ± 0.6 *

Legend: HR = heart rate; RR = respiratory rate; SAP = systolic arterial blood pressure; MAP = mean arterial blood pressure; DAP = diastolic arterial blood pressure; EtCO_2_ = end-tidal carbon dioxide tension; T °C = body temperature. The values were measured at the baseline and during premedication, anesthesia induction, mastectomy, and skin suturing. * Differences from the baseline; ^p^ global differences among all the groups; The data are expressed as mean ± SD.

**Table 2 animals-14-01902-t002:** Cumulative Pain Scores of intraoperative responses to surgical stimulus (CPS) after administration of 2 µg kg^−1^ dexmedetomidine, 4 mg kg^−1^ tramadol, 5 mg kg^−1^ tiletamine/zolazepam, and 4 mg kg^−1^ isoflurane and lidocaine in surgical incision line infiltration with traditional syringe (S) and Comfort-in device (C) compared to control (CTR) during mastectomy and suturing skin.

CPS		
^p^	Mastectomy	Suturing Skin
**CTR**	8.5 (4/11)	9 (5/10)
**S** ^α^	1.5 (0/8) *	7.5 (5/10) *
**C** ^β^	2 (0/8)	5 (1/8)

The scores were assigned according to the percent variations compared to the values recorded after general anesthesia of the heart rate, the respiratory rate, and systolic arterial blood pressure based on the following scheme: 0 ≤ 0%; 1 = variation ≤ 10%; 2 = variation > 10% but ≤20%; 3 = variation > 20%, but ≤30%; and 4 = variation > 30% or more. The total score was calculated by summing the scores. The values are expressed as the median (range). * Difference along the timeline; ^p^ global differences among all the groups; ^α^ difference between groups CTR and S; ^β^ difference between groups CTR and C; The values are expressed as the median and range.

**Table 3 animals-14-01902-t003:** The Colorado Pain Scores of postoperative analgesia were assigned from awakening, every 6 h, until 24 h after surgery (from 0 to 4 scores). Point 2 (moderate and mild pain) was the cut-off point for the administration of postoperative rescue analgesia.

CSU-CAPS					
^p^	Awakening	6 h	12 h	18 h	24 h
**Control (CTR)**	0 (0/0)	1 (0/1) *	1 (1/2) *	2 (1/2) *	2 (1/2) *
**Syringe (S)**	0 (0/0)	1 (0/1) *	1 (0/1) *	1 (1/1) *^α^	1 (1/2) *^α^
**Comfort-in (C)**	0 (0/0)	1 (0/1) *	1 (0/1) *^βγ^	1 (0/1) *^βγ^	1 (1/1) *^βγ^

* Difference along the timeline; ^p^ global differences among all the groups; ^α^ difference between groups CTR and S; ^β^ difference between groups CTR and C; ^γ^ difference between groups C and S. The values are expressed as the median and range.

**Table 4 animals-14-01902-t004:** Biochemical parameters: glycemia, aspartate transaminase (AST), alanine aminotransferase (ALT), total protein, albumin, and blood urea nitrogen (BUN) at the baseline and 12 h after surgery. The values are expressed as mean ± SD.

	CTR		S		C	
	Baseline	12 h after Surgery	Baseline	12 h after Surgery	Baseline	12 h after Surgery ^p^
**Glycemia (mmol/L)**	4.87 ± 1.3	5.03 ± 3.3 *	4.88 ± 1.8	5.41 ± 2.1 *^α^	4.77 ± 1.0	7.11 ± 1.6 *^βγ^
**Albumin (g/dL)**	5 ± 0.3	4 ± 0.2 *	4.9 ± 0.7	3.8 ± 0.1 *	5.7 ± 0.7	3.4 ± 0.1 *
**ALT (U/L)**	18 ± 2	18 ± 2.7	20 ± 0.5	21 ± 1.7	22 ± 0.2	21 ± 0.3
**AST (U/L)**	28 ± 6.2	28 ± 6.7	34 ± 0.5	34 ± 0.7	21 ± 3	21 ± 0.3
**Total Protein (g/dL)**	6.8 ± 0.3	6.8 ± 0.4	6.5 ± 0.7	6.5 ± 0.8	6.2 ± 0.2	6.2 ± 0.2
**BUN (mg/dL)**	19 ± 1.4	19 ± 1.2	21 ± 1.2	21 ± 1.3	23 ± 1.7	23 ± 1.9

* differences from the baseline; ^p^ global differences among all the groups; ^α^ difference between groups CTR and S; ^β^ difference between groups CTR and C; ^γ^ difference between groups C and S. The values are expressed as mean ± SD.

**Table 5 animals-14-01902-t005:** Malondialdehyde (MDA), catalase (CAT), superoxide dismutase (SOD), myeloperoxidase (MPO), and butyrylcholinesterase (BuChE) at baseline (B), 24 h, and 48 h after surgery in control (CTR), syringe (S) and Comfort-in (C) groups.

	B	24 h after Surgery	48 h after Surgery
**CAT (U/mL)**			
**CTR**	8.384 ± 1.472	9.561 ± 1.520 *	8.345 ± 1.069 **
**S**	8.297 ± 0.382	9.414 ± 0.296	8.553 ± 0.366
**C**	8.439 ± 0.760	9.399 ± 0.311	8.246 ± 0.293 **
**SOD (U/mL)**			
**CTR**	2.426 ± 0.320	2.526 ± 0.069	2.437 ± 0.065
**S**	2.467 ± 0.078	2.552 ± 0.293	2.411 ± 0.030
**C**	2.440 ± 0.212	2.505 ± 0.041	2.397 ± 0.050
**MPO (U/mL)**			
**CTR**	1.476 ± 0.063	1.530 ± 0.063	1.464 ± 0.070
**S**	1.466 ± 0.067	1.540 ± 0.056	1.461 ± 0.048
**C**	1.467 ± 0.099	1.512 ± 0.070	1.432 ± 0.065
**BuChe (U/mL)**			
**CTR**	0.564 ± 0.298	0.680 ± 0.301	0.627 ± 0.260
**S**	0.574 ± 0.143	0.696 ± 0.197	0.631 ± 0.189
**C**	0.572 ± 0.068	0.634 ± 0.170	0.547 ± 0.074
**MDA (mg\mL)**			
**CTR**	53.730 ± 1.495	61.210 ± 3.544 *	59.470 ± 3.724 *
**S**	53.650 ± 1.827	61.570 ± 2.817 *	58.130 ± 2.491 *^γ^
**C**	53.800 ± 1.395	55.700 ± 1.517 ^βγ^	53.580 ± 1.594 ^β^

* Differences from the baseline; ** differences between 24 h and 48 h; ^β^ differences between CTR and C; ^γ^ differences between S and C. The values are expressed as mean ± SD.

## Data Availability

The data presented in this study are available on request from the corresponding author.

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
