# Peer review of "Surgical Site Infiltration with Comfort-in Device and Traditional Syringe in Dogs Undergoing Regional Mastectomy: Evaluation of Intra- and Postoperative Pain and Oxidative Stress"

_animals, 2024, doi:10.3390/ani14131902_

Round 1

Reviewer 1 Report

Comments and Suggestions for Authors

Title: I think that pain should be the primary outcome when studying the effects of local anesthetics, and oxidative stress should be a secondary outcome. Alleviation of pain is a very obvious and clinically relevant outcome, whereas the authors have not justified their hypothesis that local anesthetics would affect oxidative stress. Therefore, pain should also be mentioned in the title before oxidative stress.

Lines 29, 118, 250, 265: 2 µg kg-1 (kg is missing; replace mc with µ)

Line 30: Do you mean IV?

Lines 35 and 36. P cannot be = 0. If your statistical software does not how values smaller than e.g. 0.001, you should write in such cases: p < 0.001.

Lines 35-36: please, revise: significantly lower

Lines 64-65: I think this sentence and its references are not needed. Removing it would also shorten the list of references, which has many such ones that are not very relevant for the main issue of this study. Please, consider removing at least such references from the list that are only referred here.

Lines 76-78: References 17-21 do not all seem to deal with the incisional needle free injection of local anesthetics used in human medicine. Please, remove all such references that are out of this focus; and if they all are, please, remove the whole sentence or look for more accurate references.

Lines 78-80: I think this sentence with its reference is not needed.

Lines 90-97: It is unclear, why you expect that the local anesthesia technique would affect the oxidative status of the dog. Please, explain it briefly.

Line 124: What was the vaporizer setting (or ETiso)? Was it stabilized? If it was adjusted, which criteria were used? Were there any significant differences in the need of isoflurane between groups?

Lines 188-207: Has this method been described earlier? If yes, please, give a reference.

Line 233: Please, say more clearly, for which analyses the data were automatically corrected by SPSS and how it was corrected.

Lines 237-238: I think this information could have told earlier, e.g.: The total sample size was of 60 subjects, and all dogs completed the study.

Lines 238-249, 275-286 and 307-337: These are quite verbose. Could it be said more briefly, e.g. No significant differences were detected in xx, yy, zz (etc.) over time nor between treatments. Then a few words about the significant differences. Please, do not repeat such results in details that can be seen in the table, but only tell the main points of them.

Line 275: This statement does not agree with the symbols in Table 3.

Lines 281-283: Some dogs needed rescue analgesia. Please, discuss if that may have affected the pain scores measured at the next time point. Was there a significant difference in the numbers of dogs needing rescue analgesia between treatments? Please, give the actual numbers for each group. Did you evaluate the duration of the local block?

Tables 1-5: It is not clear, what “Difference along the timeline” means. It would be clearer to indicate differences from baseline with “*”.

Tables 2 and 3: Please, tell in the table legend, what the numbers in the brackets mean (e.g. (0/1)). Furthermore, it is not clear, at which time points differences were detected between treatments.

Table 4: Please, use SI units.

Table 5: Please, also tell the units of the analytes. The symbols are very difficult to read, because there are so many different ones. I think that one symbol (e.g. *) is needed to indicate difference from baseline within a treatment, and three ones to indicate differences between treatments within a time point. This is indicated much clearer in Table 1. Please, also use the same system with the symbols as in Table 1 in all other tables.

The Discussion is partly unclear and sometimes out of focus. There is some repetition of results. The results should be interpreted using references to previous studies. In general, the Discussion needs major revision.

Lines 347-357: Please, do not repeat the results with the same words as in the Results section. For example, P-values should not be repeated in the discussion. Although a brief summary of the main results is presented in the beginning of the Discussion, detailed repetition should be avoided.

Lines 358-364: Please, begin the interpretation of your results with the most important ones, which I think are related to alleviation of pain. Albumin is a very secondary result, and it should be discussed much later and very briefly. Furthermore, this sentence in line 358 is quite undefinied, and it should be removed, if you cannot give references to show that such small changes in blood albumin concentration are related to patient well-being. I think the next sentence about albumin is enough in this case, and the following sentences about albumin in lines 360-364 could also be removed.

Line 366: References 40 and 42 do not seem to match this statement. Please, check them and remove the purposeless ones.

Lines 367-368: Again, please, do not repeat the results, but try to interpret them. Please, tell what this finding means or remove the sentence. However, do not over-interpret your results, if they have no obvious relevance.

Lines 372-373: You should explain clearly (with references) either here or already in the Introduction, why (with which mechanism) you think that pain and/or analgesia are associated with MDA. Please, also evaluate the magnitude of the increase; is it clinically relevant? What is considered “normal”? Are there any reference values for dogs?

Lines 374-376: This sentence is quite loose and not well justified. How does local anesthesia affect oxidative stress?

Lines 383-389: In this paragraph, you deal with both pain and oxidative stress in quite a messy way. Are they associated with each other? If yes, please, tell how; if not, please, deal them separately.

Line 397: I think this is the primary result in your study. You should discuss it earlier. Having analgesia as your primary outcome would also be in line with your conclusions.

Did the study have any limitations? If yes, they should be discussed in the end of Discussion. E.g. were the pain scores sensitive and specific enough for this kind of a study? Was the isoflurane concentration standardized and/or monitored during surgery? If no, and if the CRT group needed more isoflurane (which you do not know in that case), could it have affected the results, e.g. heart rate and blood pressure and thus also CPS, and which way?

Line 409: “showed lower inflammatory oxidative stress” is quite a strong expression considering that only one of the biomarkers was significantly lower in the Comfort-in group. Please, say it more cautiously, e.g. “suggested”. If analgesic effect was your primary outcome (as I think it was), you should deal it before oxidative stress in conclusions.

Comments on the Quality of English Language

The English language needs revision.

Author Response

title: I think that pain should be the primary outcome when studying the effects of local anesthetics, and oxidative stress should be a secondary outcome. Alleviation of pain is a very obvious and clinically relevant outcome, whereas the authors have not justified their hypothesis that local anesthetics would affect oxidative stress. Therefore, pain should also be mentioned in the title before oxidative stress.

Dear reviewer, thanks for the suggestion, we changed the title

Lines 29, 118, 250, 265: 2 µg kg-1 (kg is missing; replace mc with µ)

Done thanks

Line 30: Do you mean IV?

Done thanks

Lines 35 and 36. P cannot be = 0. If your statistical software does not how values smaller than e.g. 0.001, you should write in such cases: p < 0.001.

Done thanks

Lines 35-36: please, revise: significantly lower

Done thanks

Lines 64-65: I think this sentence and its references are not needed. Removing it would also shorten the list of references, which has many such ones that are not very relevant for the main issue of this study. Please, consider removing at least such references from the list that are only referred here.

thanks for the suggestion we have removed the references

Lines 76-78: References 17-21 do not all seem to deal with the incisional needle free injection of local anesthetics used in human medicine. Please, remove all such references that are out of this focus; and if they all are, please, remove the whole sentence or look for more accurate references.

thanks for the suggestion but there are no appropriate references, we have removed the word incisional

Lines 78-80: I think this sentence with its reference is not needed.

Thanks for the suggestion, however it is important for the application that the ago-free syringe has had in veterinary medicine.

Lines 90-97: It is unclear, why you expect that the local anesthesia technique would affect the oxidative status of the dog. Please, explain it briefly.

Thanks for the suggestion, however here we are talking about surgical techniques and anesthetic protocols in general. We have replaced the word reduce with influence

Line 124: What was the vaporizer setting (or ETiso)? Was it stabilized? If it was adjusted, which criteria were used? Were there any significant differences in the need of isoflurane between groups?

Thanks for the tip. Inspired contraction of sevoflurane (ETsevo mmHg) was recorded. We have added the details.

Lines 188-207: Has this method been described earlier? If yes, please, give a reference

Thanks for the tip. done

Line 233: Please, say more clearly, for which analyses the data were automatically corrected by SPSS and how it was corrected.

We specified

Lines 237-238: I think this information could have told earlier, e.g.: The total sample size was of 60 subjects, and all dogs completed the study.

Thanks for the tip done

Lines 238-249, 275-286 and 307-337: These are quite verbose. Could it be said more briefly, e.g. No significant differences were detected in xx, yy, zz (etc.) over time nor between treatments. Then a few words about the significant differences. Please, do not repeat such results in details that can be seen in the table, but only tell the main points of them.

Thanks for the tip done

Line 275: This statement does not agree with the symbols in Table 3.

line 275 refers to table 2

Lines 281-283: Some dogs needed rescue analgesia. Please, discuss if that may have affected the pain scores measured at the next time point. Was there a significant difference in the numbers of dogs needing rescue analgesia between treatments? Please, give the actual numbers for each group. Did you evaluate the duration of the local block?

Thank you for the valuable suggestions we have integrated into the text.

Tables 1-5: It is not clear, what “Difference along the timeline” means. It would be clearer to indicate differences from baseline with “*”.

Done thanks

Tables 2 and 3: Please, tell in the table legend, what the numbers in the brackets mean (e.g. (0/1)). Furthermore, it is not clear, at which time points differences were detected between treatments.

Thanks for the suggestion, we have added the missing information and made the corrections.

Table 4: Please, use SI units.

Thanks for the tip done

Table 5: Please, also tell the units of the analytes. The symbols are very difficult to read, because there are so many different ones. I think that one symbol (e.g. *) is needed to indicate difference from baseline within a treatment, and three ones to indicate differences between treatments within a time point. This is indicated much clearer in Table 1. Please, also use the same system with the symbols as in Table 1 in all other tables.

Thanks for the tip done

The Discussion is partly unclear and sometimes out of focus. There is some repetition of results. The results should be interpreted using references to previous studies. In general, the Discussion needs major revision.

Lines 347-357: Please, do not repeat the results with the same words as in the Results section. For example, P-values should not be repeated in the discussion. Although a brief summary of the main results is presented in the beginning of the Discussion, detailed repetition should be avoided.

Thanks for the tip done

Lines 358-364: Please, begin the interpretation of your results with the most important ones, which I think are related to alleviation of pain. Albumin is a very secondary result, and it should be discussed much later and very briefly. Furthermore, this sentence in line 358 is quite undefinied, and it should be removed, if you cannot give references to show that such small changes in blood albumin concentration are related to patient well-being. I think the next sentence about albumin is enough in this case, and the following sentences about albumin in lines 360-364 could also be removed.

Thanks for the tip done

Line 366: References 40 and 42 do not seem to match this statement. Please, check them and remove the purposeless ones.

Thanks for the tip done

Lines 367-368: Again, please, do not repeat the results, but try to interpret them. Please, tell what this finding means or remove the sentence. However, do not over-interpret your results, if they have no obvious relevance.

Thanks for the suggestion we have removed the sentence

Lines 372-373: You should explain clearly (with references) either here or already in the Introduction, why (with which mechanism) you think that pain and/or analgesia are associated with MDA. Please, also evaluate the magnitude of the increase; is it clinically relevant? What is considered “normal”? Are there any reference values for dogs?

Thanks for the tip done

Lines 374-376: This sentence is quite loose and not well justified. How does local anesthesia affect oxidative stress?

Thanks for the tip done

Lines 383-389: In this paragraph, you deal with both pain and oxidative stress in quite a messy way. Are they associated with each other? If yes, please, tell how; if not, please, deal them separately.

Thanks for the tip done

Line 397: I think this is the primary result in your study. You should discuss it earlier. Having analgesia as your primary outcome would also be in line with your conclusions.

Thanks for the tip done

Did the study have any limitations? If yes, they should be discussed in the end of Discussion. E.g. were the pain scores sensitive and specific enough for this kind of a study? Was the isoflurane concentration standardized and/or monitored during surgery? If no, and if the CRT group needed more isoflurane (which you do not know in that case), could it have affected the results, e.g. heart rate and blood pressure and thus also CPS, and which way?

Thanks for the tip done

Line 409: “showed lower inflammatory oxidative stress” is quite a strong expression considering that only one of the biomarkers was significantly lower in the Comfort-in group. Please, say it more cautiously, e.g. “suggested”. If analgesic effect was your primary outcome (as I think it was), you should deal it before oxidative stress in conclusions.

Thanks for the tip done

Reviewer 2 Report

Comments and Suggestions for Authors

Thank you for papper.

 I believe it is an important topic that is fundamental to begin evaluating the impact of analgesia on surgical oxidative stress. However, the work needs significant improvements.

The materials and methods are very unclear, and this is reflected in the rest of the work. I believe the methodology of the study needs much clarification.

The discussion needs to be much better developed in terms of debating the results and comparing them with already published information.

In the abstract - Line 29 - It would be important to indicate what the acronyms C and S refer to, as this is only indicated in the CTR. The acronyms are presented in the main document, but not in the abstract, and in my view, they should be included to facilitate reading. Lines 109-110 - Did you always take the same number and type of mammary gland? That is, the last three caudal mammary glands?

Lines 127-130 - At what distance from the tumor lesion was the infiltration performed? 1 cm from the lesion as described in the mastectomy section? How did you distribute the applications? How many applications were necessary in each case? Did you record this? This seems to me a very important point for reproducibility.

Line 138 - Was EtCO2 only evaluated after induction? This is not clear in the text. Please clarify what was evaluated at each moment.

Line 139 - Did you measure rectal temperature? Please indicate this.

Lines 144-148 - At what moments was this scale used? It seems to be indicated in the results, but not here. In any case, at what moment of the mastectomy and suturing was the scale applied? At the beginning, in the middle, or at the end of the procedure?

Lines 144-148 - It is not clear which elements (FR, RR, NIBSP) were taken into account for classification. Did all parameters need to increase by a certain percentage? Just one of them? If more than one increased by different percentages, how was the selection made? Please be clearer about the criteria.

Lines 146-148 - Is this scale original? Was any reference used? I think it could be presented in a table format. This would enrich the work.

Line 150 - Considering you used lidocaine and its duration of effect, why was it only evaluated after 6 hours?

Line 259 - Cumulative Pain Score (CPS) should have been mentioned in the Materials and Methods, and the description of how the summation was performed (as shown in Table 2) should be in the Materials and Methods. Again, I emphasize that the MM are not clear or reproducible.

Lines 281-286 - It would be important to indicate the number of animals that received analgesic rescue in each group.

Lines 346-348 - How can you state that C has greater analgesic power compared to S? In lines 276-277 you indicate that there are no differences. I think you cannot make this statement.

How do you comment on the proximity of the local anesthetic application so close to a neoplastic lesion and the possibility of the anesthetic serving as a means of neoplastic dissemination?

Line 353 - Shouldn't you explain and compare the obtained results with the bibliography in the discussion? Why didn't you do this? What is the explanation for the difference in the pain scale being observed only 18 hours after surgery?

Lines 358-360 - Add a bibliographic reference for these statements.

Line 368 - The glucose value is a very nonspecific value and can be increased for many other reasons. Please indicate that it is not a parameter with increased specificity.

Move lines 374 to line 369, that is, before discussing oxidative indicators.

Line 377 - Please indicate that this is in humans. Unfortunately, in veterinary medicine, we do not know if it is the same!

Lines 383-389 - I suggest eliminating this section; it is not directly related to the results or adds anything to the focused work.

Lines 403-404 - I suggest eliminating this sentence: "This feature makes the Comfort-in device suitable for dental extractions, urology, diabetology, virology, dermatology, and pediatrics [18,20,21].”

Line 408 - I think the conclusion cannot be this. There are no differences in the pain scale or in the stress and inflammation markers between groups C and S. Therefore, the conclusion needs to be rewritten.

Comments on the Quality of English Language

minor edition

Author Response

 I believe it is an important topic that is fundamental to begin evaluating the impact of analgesia on surgical oxidative stress. However, the work needs significant improvements.

The materials and methods are very unclear, and this is reflected in the rest of the work. I believe the methodology of the study needs much clarification.

The discussion needs to be much better developed in terms of debating the results and comparing them with already published information.

In the abstract - Line 29 - It would be important to indicate what the acronyms C and S refer to, as this is only indicated in the CTR. The acronyms are presented in the main document, but not in the abstract, and in my view, they should be included to facilitate reading.

Done thanks

Lines 109-110 - Did you always take the same number and type of mammary gland? That is, the last three caudal mammary glands?

Thanks for the suggestions we have specified.

Lines 127-130 - At what distance from the tumor lesion was the infiltration performed? 1 cm from the lesion as described in the mastectomy section? How did you distribute the applications? How many applications were necessary in each case? Did you record this? This seems to me a very important point for reproducibility.

Thanks for the suggestions we have specified.

Line 138 - Was EtCO2 only evaluated after induction? This is not clear in the text. Please clarify what was evaluated at each moment.

Done thanks

Line 139 - Did you measure rectal temperature? Please indicate this.

Done thanks

Lines 144-148 - At what moments was this scale used? It seems to be indicated in the results, but not here. In any case, at what moment of the mastectomy and suturing was the scale applied? At the beginning, in the middle, or at the end of the procedure?

Done thanks

Lines 144-148 - It is not clear which elements (FR, RR, NIBSP) were taken into account for classification. Did all parameters need to increase by a certain percentage? Just one of them? If more than one increased by different percentages, how was the selection made? Please be clearer about the criteria.

Done thanks

Lines 146-148 - Is this scale original? Was any reference used? I think it could be presented in a table format. This would enrich the work.

Done thanks

Line 150 - Considering you used lidocaine and its duration of effect, why was it only evaluated after 6 hours?

Thanks for the observation. The evaluation was performed three hours after administration of lidocaine for all dogs and corresponded to the animal being completely awake. We have specified  in the text and in the table

Line 259 - Cumulative Pain Score (CPS) should have been mentioned in the Materials and Methods, and the description of how the summation was performed (as shown in Table 2) should be in the Materials and Methods. Again, I emphasize that the MM are not clear or reproducible.

Thanks for the observation We have specified the summation was performed for  obtained CPS  in MM

Lines 281-286 - It would be important to indicate the number of animals that received analgesic rescue in each group.

Thanks for the observation We have specified the number of animals that received analgesic rescue in each group.

Lines 346-348 - How can you state that C has greater analgesic power compared to S? In lines 276-277 you indicate that there are no differences. I think you cannot make this statement.

How do you comment on the proximity of the local anesthetic application so close to a neoplastic lesion and the possibility of the anesthetic serving as a means of neoplastic dissemination?

 Thanks for the observation . The subjects in group C did not require rescue analgesia either intra or post-operatively, while in the other two groups, some subjects required the administration of rescue analgesic. This could have influenced the scores. Local anesthesia with lidocaine blocks the activation of prometastatic pathways. We have specified

Line 353 - Shouldn't you explain and compare the obtained results with the bibliography in the discussion? Why didn't you do this? What is the explanation for the difference in the pain scale being observed only 18 hours after surgery?

Thanks for the observation done. Patients were monitored for analgesia for only 24 hours as that is how long they were hospitalized after surgery. This represented a further limitation of the study

Lines 358-360 - Add a bibliographic reference for these statements.

Thanks for the observation done

Line 368 - The glucose value is a very nonspecific value and can be increased for many other reasons. Please indicate that it is not a parameter with increased specificity.

Thanks for the observation done

Move lines 374 to line 369, that is, before discussing oxidative indicators.

Thanks for suggestion done

Line 377 - Please indicate that this is in humans. Unfortunately, in veterinary medicine, we do not know if it is the same!

There are also studies in animals and dogs.

Lines 383-389 - I suggest eliminating this section; it is not directly related to the results or adds anything to the focused work.

Thanks for the observation, but Pain, and its consequences on homeostasis, is the focused work

Lines 403-404 - I suggest eliminating this sentence: "This feature makes the Comfort-in device suitable for dental extractions, urology, diabetology, virology, dermatology, and pediatrics [18,20,21].”

Thanks for suggestion done

Line 408 - I think the conclusion cannot be this. There are no differences in the pain scale or in the stress and inflammation markers between groups C and S. Therefore, the conclusion needs to be rewritten.

Thanks for suggestion done

Round 2

Reviewer 2 Report

Comments and Suggestions for Authors

Thank you for the improvements. The document is significantly better and, in my opinion, meets the requirements for publication.